# Silencing *Autophagy-Related Gene 2* (*ATG2*) Results in Accelerated Senescence and Enhanced Immunity in Soybean

**DOI:** 10.3390/ijms222111749

**Published:** 2021-10-29

**Authors:** Said M. Hashimi, Ni-Ni Wu, Jie Ran, Jian-Zhong Liu

**Affiliations:** 1Institute of Plant Genetics and Developmental Biology, College of Chemistry and Life Sciences, Zhejiang Normal University, Jinhua 321004, China; s.masoud.hashimi@gmail.com (S.M.H.); wunini23@163.com (N.-N.W.); ranjie15736516243@163.com (J.R.); 2Zhejiang Provincial Key Laboratory of Biotechnology on Specialty Economic Plants, Zhejiang Normal University, Jinhua 321004, China

**Keywords:** *Glycine max*, autophagy, virus-induced gene silencing (VIGS), senescence, ROS, salicylic acid, immune response, *Gm*MPK3/6

## Abstract

Autophagy plays a critical role in nutrient recycling and stress adaptations. However, the role of autophagy has not been extensively investigated in crop plants. In this study, soybean *autophagy-related gene 2* (*GmATG2*) was silenced, using virus-induced silencing (VIGS) mediated by *Bean pod mottle virus* (BPMV). An accelerated senescence phenotype was exclusively observed for the *GmATG2-*silenced plants under dark conditions. In addition, significantly increased accumulation of both ROS and SA as well as a significantly induced expression of the pathogenesis-related gene 1 (*PR1*) were also observed on the leaves of the *GmATG2-*silenced plants, indicating an activated immune response. Consistent with this, *GmATG2*-silenced plants exhibited a significantly enhanced resistance to *Pseudomonas syringae* pv. *glycinea* (Psg) relative to empty vector control plants (BPMV-0). Notably, the activated immunity of the *GmATG2-*silenced plants was independent of the MAPK signaling pathway. The fact that the accumulation levels of ATG8 protein and poly-ubiquitinated proteins were significantly increased in the dark-treated *GmATG2-*silenced plants relative to the BPMV-0 plants indicated that the autophagic degradation is compromised in the *GmATG2-*silenced plants. Together, our results indicated that silencing *GmATG2* compromises the autophagy pathway, and the autophagy pathway is conserved in different plant species.

## 1. Introduction

Autophagy is a fundamental recycling intracellular constituent pathway among eukaryotes, including yeast, *C. elegans*, *Drosophila melanogaster*, humans and plants [1,2]. This process is essential for plants’ adaptation to a variety of environmental stresses, including nutrient deprivation, oxidative stress, salt, drought, and pathogen attack [1,2]. The central process of autophagy is fulfilled by a group of autophagy-related genes (ATGs) that were originally characterized and investigated in yeast [2]. Around 40 ATG genes have been characterized in yeast, and most of them have homologs in plants [1,2], suggesting the conservation of the core ATG process during evolution. In Arabidopsis, most of the *ATG* genes have more than one isoform, revealing functional redundancy [1,2]. 

Autophagy involves the engulfing of cytoplasmic components in the autophagosomes, which subsequently fuse with the lysosomes (in animals) or vacuoles (in yeast and plants) and result in degradation of their cargoes by various hydrolases [1,2,3,4]. Cargo breakdown results in the production of molecular building blocks, such as amino acids, which are then released back into the cytoplasm for reuse [1,2,3]. Autophagy could be non-selective or selective [1,2,3,4]. Non-selective autophagy, also referred to as bulk autophagy, uptakes and degrades the cytosolic compounds non-selectively in response to nutrient starvation, whereas selective autophagy mediates the degradation of target compounds by specific cargo receptors, which specifically interact with autophagosome membrane-anchored ubiquitin-like ATG8 proteins [5] via an ATG8-interacting motif (AIM) with a core W/Y/F-XX-L/I/V sequence [6,7,8]. It has been demonstrated that the selective autophagy plays central roles in removing protein aggregates [9,10,11,12] and damaged organelles [13,14,15,16,17,18,19] under stress conditions. In addition, selective autophagy also plays a crucial role in the clearance of invading pathogens, such as bacteria, viruses, and fungi [20,21,22,23,24,25]. 

Autophagy has been shown to play both pro- and anti-cell death functions [9,26,27,28,29]. Beclin1/ATG6-silencing results in unrestricted cell death beyond the sites of TMV infection on the leaves of *N. benthamiana* plants expressing the *N* gene [26], whereas the loss function of *ATG* genes leads to compromised RPS4- or RPM1-triggered HR [27]. *atg* mutants in Arabidopsis often display accelerated senescence or cell death phenotypes when in nutrient-deficient conditions, or even under nutrient-rich conditions [28,30,31,32,33,34,35,36,37]. The early senescence/cell death in *atg* mutant plants is dependent on biosynthesis and the over-accumulation of both ROS and SA [28,33]. 

ATG2 encodes a key autophagy-related protein that plays a critical role in assisting ATG9 protein to deliver lipids to the expanding phagophore together with ATG18 protein [1,2]. Yeast ATG2 localizes to the pre-autophagosomal structure (PAS) and forms a complex with ATG18, which directly associates with phosphatidylinositol 3-phosphate (PtdIns(3)P) [38]. The localization of ATG2 to PAS is dependent on the binding of ATG18 to PtdIns(3)P [39], and the localization of the ATG2-ATG18 complex to the PAS requires the phosphatidylinositol 3-kinase (PI3K) complex [38]. The loss function of ATG2 in Arabidopsis inhibits autophagy initiation or autophagosome formation, thus leading to autophagy defective phenotypes, such as hypersensitivity to nutrient starvation, accelerated senescence, cell death and enhanced resistance to biotrophic pathogens, which depends on the ROS and SA pathway [28,33].

Soybean (*Glycine max*) is one of the most important crops worldwide. Soybean seeds are rich sources of vegetable oil and proteins. Pathogen infections cause significant economic losses in soybean production (http://aes.missouri.edu/delta/research/soyloss.stm accessed on 24 October 2021). Therefore, understanding immunity signaling pathways is fundamental to improve disease resistance via transgenic approaches in soybean. The autophagy pathway is critical in regulating plant immunity and nutrient recycling. Although extensively studied in model plant Arabidopsis, it has rarely been investigated in soybean. In this study, we aimed to understand the roles of autophagy in soybean immunity by silencing *GmATG2*. We showed that silencing *GmATG2* led to an accelerated senescence under dark conditions. Similar to *atg* mutants in Arabidopsis, ROS- and SA-dependent immune responses and enhanced resistance to biotrophic pathogen were observed in the *GmATG2-*silenced plants. Interestingly, the activated immunity of the *GmATG2-*silenced plants was independent of the MAPK signaling pathway. Furthermore, we showed that the autophagic degradation was compromised in the dark-treated *GmATG2-*silenced plants. Collectively, our results indicated that silencing *GmATG2* impairs autophagy pathway, and autophagy pathway is conserved across plant species. 

## 2. Results

### 2.1. Silencing Autophagy-Related Gene 2 (GmATG2) in Soybean Results in Accelerated Senescence Phenotypes under Dark Conditions 

Autophagy has been well studied in model plant Arabidopsis. However, this important process has not been extensively investigated in crops, such as soybean. To understand the roles of autophagy in soybean, we determined to silence one of the *ATG* genes, *GmATG2*, in soybean using *Bean pod mottle virus* (BPMV)–induced gene silencing (BPMV-VIGS) [40,41,42]. Soybean is a paleotetraploidy, with around 75% of genes existing in multiple copies in its genome [43]. We found that *GmATG2* has two copies in the soybean genome (Glyma.02G133400 and Glyma.07G211600, referred to as *GmATG2a* and *GmATG2b*, respectively), sharing an identity above 96% at the nucleotide level. A blast search revealed that the two paralogs of *GmATG2* share an identity of >60% at the nucleotide level with the Arabidopsis counterparts (Table 1). The BPMV-VIGS system is currently one of the most powerful tools for interrogating gene function in soybean and has been shown to be effective in resolving the issue of gene redundancy in soybean [44,45]. To silence *GmATG2*, we cloned a fragment of about 330-bp into BPMV2 vector [40]. Because the 330-bp fragment shares >95% homology within the two *GmATG2* paralogs, we expect that both paralogs will be silenced simultaneously, as it has been shown previously for other paralogous genes in soybean [46,47,48,49,50]. The plasmid of the constructed *BPMV2-GmATG2* was then co-delivered into 7-day-old soybean seedlings, together with *BPMV1* plasmid via biolistic bombardments, which would result in infection of the soybean seedlings and thus, induction of silencing. After successful infection, viral symptoms were visible in about 15–20 days post bombardment on the systemic leaves. These symptomatic leaves were then collected, and the viral saps were extracted by grinding the leaves in the phosphate buffer for subsequent inoculation in large scales simply by rubbing the first true leaves of the soybean seedlings. No significant difference in phenotype was observed between the BPMV-0 plants and the *GmATG2-*silenced plants (Figure 1A,B), indicating that silencing *GmATG2* has no dramatic effects on the growth and development of soybean under normal conditions. RT-PCR analysis showed that *GmATG2* was successfully silenced because the transcript level was significantly reduced in the *GmATG2*-silenced plants relative to the BPMV-0 plants (Figure 1C). At the same time, we also used primers on the BPMV2 vector flanking the inserted fragments to conduct the RT-PCR analysis. The sizes of the amplified fragments from the silenced plants were larger than the fragments amplified from the BPMV-0 plants (Figure 1D), indicating that the BPMV carrying the target fragments successfully infected the soybean plants.

### 2.2. Accelerated Senescence Was Observed in the GmATG2-Silenced Plants under Dark Treatment

One of the signature characteristics of the *ATG* mutants in Arabidopsis is their significantly elevated sensitivity to nutrient starvation [1,2,13]. Plants grown under a continuous dark condition suffer from carbon deficiency because of the lack of photosynthesis [51]. Autophagy plays a key role for breaking down the damaged organelles or other cellular components and replenishing the carbon source for cells under dark conditions [52]. Because no visible phenotypes were observed for the *GmATG2*-silenced plants in comparison with the BPMV-0 plants, we decided to treat these plants under dark. The detached leaves from both the BPMV-0 and the *GmATG2*-silenced plants were put onto a moisturized filter paper in a petri dish and then incubated in a growth chamber with the lights off. After 7 days of dark treatment, we found that while the BPMV-0 leaves were still green and healthy, the *GmATG2*-silenced leaves clearly displayed a yellowish senescence phenotype (Figure 2), indicating that the autophagy pathway was impaired in the *GmATG2*-silenced plants.

### 2.3. The Accumulation Level of the ATG8 Protein Enhanced in the GmATG2-Silenced Plants under Dark Conditions

Over 30 ATG genes have been identified in plants, which are involved in different steps of autophagy pathway [1,2,53]. ATG8 ubiquitin-like proteins play multiple roles in the autophagy pathway, which present on the membrane of the autophagosomes and are ultimately degraded in the vacuoles [1,2,3]. If the autophagy pathway is impaired, the accumulation level of ATG8 proteins is elevated. To examine whether the autophagy pathway is impaired in the *GmATG2*-silenced plants, Western blotting analysis was performed for the protein samples extracted from the dark-treated leaves of both the BPMV-0 and *GmATG2*-silenced plants, using an antibody raised against the Arabidopsis ATG8 (AS14 2769, Agrisera), which can cross-react with the ATG8s from soybean. The Western blotting analysis result showed that the accumulation level of the *Gm*ATG8 proteins was significantly higher in the leaves of the *GmATG2*-silenced plants than in the leaves of the BPMV-0 plants (Figure 3A), indicating that the silencing of *GmATG2* impaired the autophagic degradation process and that the function of *Gm*ATG2 is critical for autophagy in soybean.

### 2.4. The Poly-Ubiquitinated Proteins Were Over-Accumulated in the GmATG2-Silenced Plants

The degradation of damaged proteins or protein aggregates resulted from various stress conditions via the autophagy pathway is an essential mechanism for the survival of organisms under the stress conditions [9,10,11,12]. The damaged proteins or protein aggregates targeted for autophagic degradation are usually polyubiquitinated. If the autophagic degradation is impaired, the accumulation level of the polyubiquitinated proteins increases, which was observed in many *atg* mutants [9,10,11,12]. In order to check whether silencing *GmATG2* affects the autophagic degradation of poly-ubiquitinated proteins, we performed Western blotting on the protein samples extracted from the dark-treated leaves of both the BPMV-0 and *GmATG2-*silenced plants, using an antibody raised against Arabidopsis ubiquitin protein (AS08307, Agrisera). The Western blotting analysis showed that the level of poly-ubiquitinated proteins was significantly increased in *GmATG2-*silenced plants relative to the vector control plants (Figure 3B), indicating that clearance of the poly-ubiquitinated proteins via the autophagy pathway is compromised in the *GmATG2*-silenced plants. 

### 2.5. Silencing GmATG2 Leads to Increased Accumulation of H_2_O_2_

The reactive oxygen species (ROS) is one of the most critical signal molecules in regulating plant immunity [54,55]. ROS accumulates in great amounts during diverse environmental stress conditions. ROS can cause damages to carbohydrates, lipids, DNA, and proteins, ultimately leading to cell death [56]. Autophagy is induced in response to ROS accumulation to decrease oxidative damage in plants cells [33]. It was shown in Arabidopsis that elevated ROS accumulation was observed in the leaves of *atg* mutants, even under normal conditions [28,33]. To examine whether a similarly enhanced ROS accumulation can be observed in the leaves of *GmATG2*-silenced plants, we performed DAB staining to visualize the accumulation level of H_2_O_2_. Consistent with the result from Arabidopsis, the level of H_2_O_2_ accumulation on the leaves of BPMV-*GmATG2*, plants was significantly increased, compared to BPMV-0 (Figure 4A,B).

### 2.6. Silencing GmATG2 Results in Increased Accumulation of SA and Highly Induced Expression of PR1 under Dark Treatment 

Salicylic acid (SA) is an essential phytohormone that regulates cell death and defense responses against biotrophic pathogens. Pathogen-inducible SA is mainly synthesized via the isochorismate (ICS1) that takes place in the chloroplast [57]. SA is responsible for regulating the expression of the pathogenesis-related (*PR*) genes in plants [57]. It was shown that the SA level is enhanced in the *atg* mutants relative to Col-0 [28,33]. In addition, the transcript level of the *PR1* was significantly induced in *atg* mutants [28,33]. To examine whether silencing *GmATG2* in soybean affects the SA accumulation and the *PR1* induction, the SA level was quantitated for the leaf samples collected from the BPMV-0 and the BPMV-*GmATG2* plants after 24 h dark treatment. As shown in Figure 4C, the bound SA level was significantly higher in the BPMV-*GmATG2* plants than in the BPMV-0 plants. For an unknown reason, we were not successful in quantitating the free SA levels in these plants. RT-PCR result showed that the expression of the *PR1* was highly induced in the BPMV-*GmATG2* plants relative to the BPMV-0 plants in response to dark treatment (Figure 4D). Collectively, these results indicated that autophagy negatively controls the SA level and *PR* gene expression in the soybean.

### 2.7. Silencing GmATG2 Results in Enhanced Resistance to Pseudomonas Syringae pv. Glycinea (Psg) 

It was shown that the autophagy pathway plays a positive role in resistance to necrotrophic pathogens [58,59]. However, comprehensive research using mutants of several *ATG* genes in Arabidopsis indicated that autophagy negatively regulates plant basal immunity against biotrophic pathogens [28,33]. In order to examine the effects of *GmATG2*-silencing on resistance against biotrophic pathogens, Psg was spray-inoculated on the leaves of both the BPMV-0 and the BPMV-*GmATG2* plants. At different day-post-spraying (dps), the leaf discs were collected and used for determining colony forming unit (cfu). As shown in Figure 5, the multiplication of Psg was significantly lower on the leaves of the BPMV-*GmATG2* plants than on the leaves of the BPMV-0 plants, indicating that silencing *GmATG2* significantly enhanced soybean resistance to Psg. 

### 2.8. The Activated Immunity Observed in the GmATG2-Silenced Plants Is Not Correlated with the Activated MAPK Signaling Pathway

flg22 is a conserved peptide derived from the N-terminal 22 amino acids of the flagellin of *Pseudomonas syringae* [60], which can be recognized by the plant receptor like kinase, FLS2 and activate the defense responses. Mitogen-activated protein kinase (MAPKs) cascade pathway acts downstream of the recognition of PAMPs by receptor-like kinases, such as FLS2, to transduce the extracellular signals into an intracellular adaptive response [61]. Previous research proposed that the MAPK signaling pathway is not affected by loss of *ATG* genes in Arabidopsis [59]. To examine the effects of *GmATG2* silencing on the activation of MAPKs in soybean, we treated the BPMV-0 and BPMV-*GmATG2* plants with 10 µM flg22 for 0–6 h followed by the Western blotting analysis, using the Phospho-p44/42 MAP Erk1/2 antibody, which specifically recognizes the phosphorylated MPK3/4/6 from different plant species. As shown in Figure 6, the activation level of *Gm*MPK3 and *Gm*MPK6 induced by flg22 significantly reduced in the BPMV-*GmATG2* plants relative to the BPMV-0 plants, indicating that *Gm*ATG2 plays a positive role in regulating the activation of *Gm*MPK3/6 in soybean. The enhanced resistance of the *GmATG2*-silenced plants against Psg is not correlated with the activation of the MAPK signaling pathway. 

## 3. Discussion

Autophagy is a recycling process that is unique in eukaryotic organisms [1,2,3]. In plants, autophagy has multiple functions in development and stress adaptations [1,2,3,4]. In Arabidopsis, many *atg* mutants exhibit an accelerated senescence and autoimmune phenotypes under natural growth conditions because they lack functional autophagy [9,20,28,33,35,59]. However, the similar phenotypes were not seen for the *GmATG2*-silenced plants under natural growth conditions (Figure 1A,B). The accelerated senescence and autoimmune phenotypes were only observed on the silenced plants that were subjected to dark treatment (Figure 2). This could be explained by the fact that the BPMV-mediated silencing could only result in reductions (80–90%) in the transcript level rather than complete loss functions of the silenced genes [44,45]. The remaining 10–20% *Gm*ATG2 in *GmATG2*-silenced plants is sufficient to maintain functional levels of autophagy under normal growth conditions. However, because photosynthesis halts under dark conditions, dark treatment for a longer period of time mimics carbon starvation, which induces significant autophagy. Under such conditions, 10–20% of normal level of *Gm*ATG2 in the *GmATG2*-silenced plants could not assemble enough autophagosomes to deal with the situation and thus, displayed the accelerated senescence and autoimmune phenotypes (Figure 2). Under natural conditions, the accelerated senescence and autoimmune phenotypes observed in Arabidopsis *atg* mutant could be resulted from complete loss of a functional autophagy pathway [28,33,35].

ATG8s play critical roles in autophagosome formation and present on the autophagosomes during the entire autophagy pathway [1,2,3]. The ATG8s are destined and degraded in the vacuoles. In Arabidopsis *atg* mutants, the level of ATG8s accumulation is elevated because of the impaired autophagic degradation [1,2]. Similar to the *atg* mutant in Arabidopsis, the accumulation level of the *Gm*ATG8 was significantly increased in the dark-treated *GmATG2*-silenced plants (Figure 3A), indicating that the autophagy pathway is impaired in the *GmATG2*-silenced plants. 

Under various stress conditions, the poly-ubiquitinated proteins or proteins aggregates formed under these conditions are eliminated through autophagic degradation pathway, and the ubiquitinated proteins are over-accumulated in many Arabidopsis *atg* mutants [9,10,11,12]. It is the inappropriate over-accumulation of ubiquitinated protein aggregates and the subsequent excessive ER stress that cause the early death of the *atg* mutants [9]. Disruption of cellular homeostasis in *atg* mutants results in SA buildup and NPR1-dependent accumulation of defense-related transcripts in a deleterious cycle [9]. Consistent with the results in Arabidopsis, the ubiquitinated proteins were over-accumulated in dark-treated *GmATG2-*silenced plants (Figure 3B), indicating that autophagic degradation of ubiquitinated proteins was compromised in the *GmATG2-*silenced plants. It remains to be determined whether the over-accumulation of ubiquitinated proteins in dark-treated *GmATG2-*silenced plants is dependent on NPR1 or the SA pathway.

We also tested the accumulation level of the other autophagy-related protein, NBR1, which contains a UBA domain and serves as a cargo receptor that targets ubiquitinated proteins or protein aggregates for autophagic degradation [10,12]. NBR1 binds to autophagosomal-bound ATG8, which is also destined and degraded in the vacuoles [1,2]. Unfortunately, because the Arabidopsis NBR1 antibody we used did not cross-react with the soybeam *Gm*NBR1, we could not draw any conclusion from this study. More experiments, such as examining the formation of autophagosomes by using a GFP-*Gm*ATG8 fusion protein, are required to draw a firm conclusion as to whether the autophagy pathway is indeed compromised in the *GmATG2-*silenced plants. Nonetheless, our results suggested that the autophagy pathway is conserved between soybean and Arabidopsis.

One of the predominant roles of autophagy is to remove ROS by the degradation of oxidized materials and organelles in the cells [62]. The elevated ROS activates defense responses and causes hypersensitive cell death in plants [54,55]. Under natural conditions, the accelerated senescence and autoimmune phenotypes observed in the Arabidopsis *atg* mutants, including the enhanced accumulation of both H_2_O_2_ and SA, the induced expression of *PR* genes, and the elevated resistance to biotrophic pathogens, are wholly dependent on the SA biosynthesis or signaling because loss function of EDS1 or PAD4 or NPR1 can rescue the accelerated senescence and autoimmune phenotypes [28,33]. Similarly, we found that the levels of both H_2_O_2_ and bound SA were significantly increased, the expression of *PR* genes was significantly induced and the resistance to *Psg* was significantly enhanced in the *GmATG2*-silenced plants, relative to the BPMV-0 plant (Figure 4 and Figure 5). These results strongly indicated that the autophagy pathway plays a negative role in regulating immunity both in Arabidopsis and soybean. However, it is unknown whether these phenotypes are dependent on the SA biosynthesis or signaling in the *GmATG2*-silenced plants. We are interested in testing whether co-silencing *GmATG2* together with *GmEDS1*, *GmPAD5*, *GmICS1* or *GmNPR1* simultaneously could rescue the autoimmune phenotypes observed in the dark-treated *GmATG2*-silenced plants. 

Upon the recognition of flg22, a fragment of bacterial flagellin, by the FLAGELLIN SENSITIVE 2 (FLS2) receptor [61], induces the hetero-dimer formation of FLS2 with its co-receptor, BRI1-ASSOCIATED KINASE 1 (BAK1) [63,64] and subsequently triggers PTI (PAMP-triggered immunity), including the activation of the MAPK cascade [65]. Similar to Arabidopsis, silencing *GmFLS2* reduced the activation of *Gm*MPK6 in response to flg22 treatment [50]. In Arabidopsis *atg* mutant plants, the activation of the MPK3/4/6 in response to elicitation was not different from that of the Col-0 plants [59]. Interestingly, it was found that the activation of the *Gm*MPK3/6 in response to flg22 treatment was significantly reduced in the *GmATG2*-silenced plants (Figure 6), indicating that the accelerated senescence and autoimmune phenotypes are independent of the MAPK signaling pathway. The reason why the activation of the *Gm*MAPKs in response to flg22 is reduced in the *GmATG2*-silenced plants is not understood.

Soybean is a paleotetraploidy; ~75% of genes in its genome have two copies [43]. Consistent with this analysis, we found two parologs for *GmATG2*. It was established that VIGS can co-silence genes sharing > 85% identity at the nucleotide level [44]. As the 330 bp fragments used for silencing *GmATG2* sharing >95% identity within the two parologous genes, we believed that the two parologous *GmATG2a* and *GmATG2b* were silenced simultaneously. The autoimmune phenotypes observed on the *GmATG2*-silenced plants were actually resulted from co-silencing two parologous genes. These results indicated once more that the BPMV-VIGS system is a power tool in resolving gene redundancy and dissecting gene functions in the paleotetraploidy soybean. 

## 4. Materials and Methods

### 4.1. Plant Materials

Soybean (*Glycine max*, Williams 82) was used in this study. Soybean plants were maintained in the growth chamber at 22 °C with a photoperiod of 16 h light/8 h dark, unless otherwise indicated. 

### 4.2. BPMV-Mediated VIGS

The BPMV-VIGS system, and inoculation of soybean seedlings with DNA-based BPMV constructs via biolistic particle bombardment using a Biolistic PDS-1000/He system (Bio-Rad Laboratories, Hercules, CA, USA) were described previously [40,41,42]. The orthologs of *GmATG2* were identified by BLAST search, using Arabidopsis *ATG2* in the Phytozome database (www.phytozome.org, accessed on 24 October 2021). A 330 bp fragment of *GmATG2a* was amplified using the following primers: *GmATG2a*-F (Glyma.02G133400).

5′-aag**GGATCC**CACATTCTACATCAGATGCTGAAT

3′, and *GmATG2a*-R: 5′-ttg**GGTACC**CAGAAGATAATGCATTAGAACCTCCT-3′; The bold letters represent BamHI and and KpnI restriction sites, respectively.

### 4.3. Inoculation of Pseudomonas syringae pv. glycinea (Psg) R4 Strain onto the Leaves of Soybean Plants 

Psg growth assay was performed as described [50]. Psg R4 was cultivated at 28 °C for about 36 h until OD_600_ = 1.3. The bacterial culture was centrifuged at 3000× *g* rpm for 10 min and the pellet was re-suspended to an OD_600_ of 1. The vector control or *GmATG2*-silenced plants were sprayed with the re-suspended bacterial solution thoroughly on both sides of the leaves. The inoculated plants were then covered by plastic bags for at least 24 h. To retain moisture, the inoculated plants were sprayed with water a few times during daytime.

### 4.4. MAPK Activity Assay

Protein was extracted from flg22-treated soybean leaf tissues in an extraction buffer (50 mm Tris-MES, pH 8, 0.5 m Suc, 1 mm MgCl_2_, 10 mm EDTA, and 5 mm DTT as described [49]. For Western blotting, proteins were separated by SDS-PAGE (10% acrylamide gel) and transferred to PVDF membranes (Millipore, Burlington, MA, USA) by semidry electrotransfer (Bio-Rad) followed by incubation with the anti-phospho-p44/p42 MAPK (anti-pTEpY) diluted at 1:2000 (Cell Signaling Technology, Danvers, MA, USA). The bands were visualized using horseradish peroxidase (HRP) substrate (Millipore). Coomassie Blue–stained gel (CBS) was used as an equal loading control.

### 4.5. Histochemical Assays

The presence of H_2_O_2_ was visualized by the 3,3-diaminobenzidine tetrahydrochloride (DAB) staining (Sigma-Aldrich, St. Louis, MO, USA) [66]. The detached leaves were placed in a solution containing 1 mg mL^−1^ DAB (pH 5.5) for 2 h first and cleared subsequently by boiling in 96% ethanol for 10 min.

### 4.6. SA Quantification

SA was quantified using an Agilent 1260 HPLC system (Agilent Technologies, Santa Clara, CA, USA) with a diode array detector and a fluorescence detector and a column as described previously [67]. 

### 4.7. RNA Isolation and RT-PCR

RNA isolation and RT-PCR were performed as described [48]. The primers used for RT-PCRs are as follows: 

*GmATG2a/2b*-F: GATATTGGAGGAAGGGGGTAT

*GmATG2a/2b*-R: TTGGCATACTAGTATCACCTGCAT

*GmELF1b-F*: ACCGAAGAGGGCATCAAATCCC

*GmELF1b-R:* CTCAACTGTCAAGCGTTCCTC

*GmPR1-F:* ATGGGGTTGTGCAAGGTT

*GmPR1-R:* CTAGTAGGGTCTTTGGCCAA

*1548-F:* CAAGAGAAAGATTTGTTGGAGGGA

*BPMV-RNA2-R:* ACAAGGAAATCCGGTACGCTT 

### 4.8. Western Blotting Analysis

Proteins were prepared from leaves collected from BPMV-0 and BPMV-*GmATG2* plants treated in the dark for different time. The immunoblots were performed as previously described [68]. Anti-Ubq (Agrisera, AS08307) and anti-ATG8 (Agrisera, AS142769) were used for Western blotting. 

## Figures and Tables

**Figure 1 ijms-22-11749-f001:**
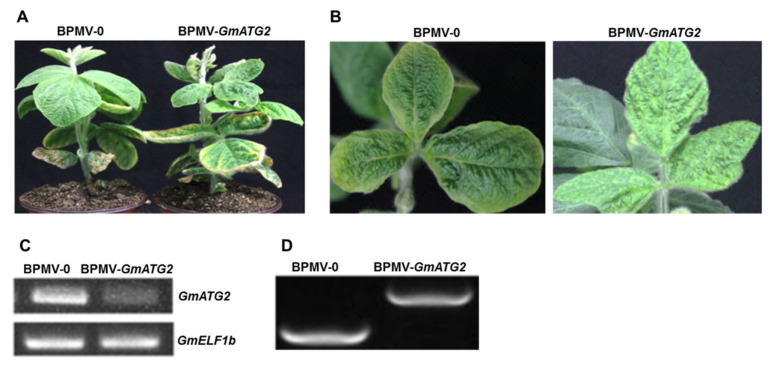
Phenotypes of silenced and control plants under normal conditions. (**A**) BPMV-0 and BPMV-*GmATG2* plants shared a similar phenotype; (**B**) the BPMV-*GmATG2* plants were not affected in growth and development, compared to BPMV-0. The leaves of the *GmATG2-*silenced plants and vector plants displayed obvious virus symptoms; there were bubbly protrusions around the veins, and the leaves are curled irregularly; (**C**) the transcript level of *GmATG2* was significantly reduced in the silenced plants, compared to the BPMV-0 plants. *GmELF1b* is used as the internal reference gene. (**D**) RT-PCR fragment amplified from the BPMV-*GmATG2*-silenced plants is larger than that amplified from the BPMV-0 plants, using a pair of primers on the BPMV2 vector. The increased length of the amplified fragment in the *GmATG2-*silenced plants is equal to the size of the inserted target gene fragment. The experiments were repeated at least twice, with similar results. At least 3 plants were used in each repeat.

**Figure 2 ijms-22-11749-f002:**
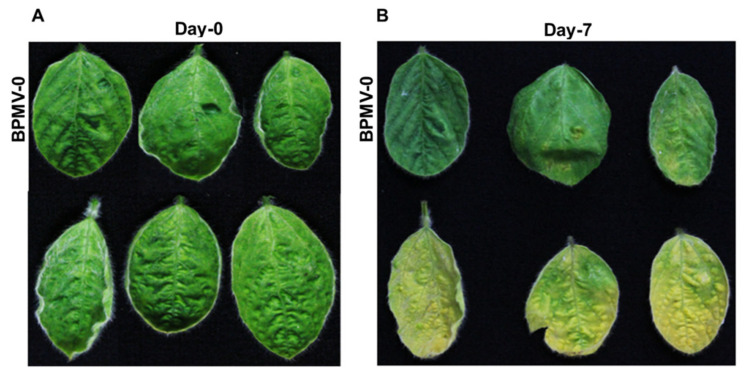
An accelerated senescence phenotype was observed on the leaves of the *GmATG2*-silenced plants treated under dark. (**A**) Comparison of the BPMV-0 leaves with the BPMV-*GmATG2* leaves before dark treatment; (**B**) comparison of BPMV-0 leaves with the BPMV-*GmATG2* leaves treated under dark for 7 days. This experiment was repeated 4 times with similar result. At least 3 plants were used in each repeat.

**Figure 3 ijms-22-11749-f003:**
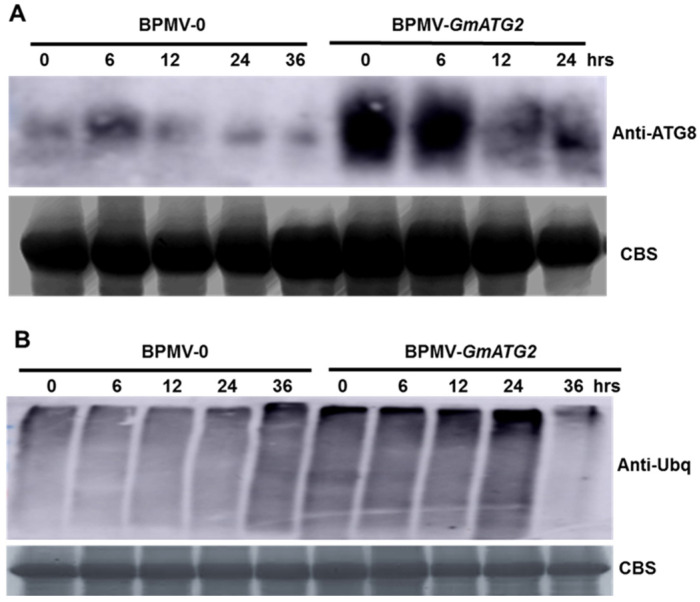
Silencing of *GmATG2* impaired autophagic degradation. (**A**) Comparison of the accumulation level of the ATG8 protein between the BPMV-0 and *GmATG2-* silenced plants; (**B**) comparison of the level of the ubiquitinated proteins between BPMV-0 and BPMV-*GmATG2* plants. Protein samples were extracted from the BPMV-0 and BPMV-*GmATG2* plants, respectively, at 0 min, 6 h, 12 h, 24 h and 36 h post dark treatment. The experiments were repeated at least 2 times with similar results. At least 3 plants were used in each iteration.

**Figure 4 ijms-22-11749-f004:**
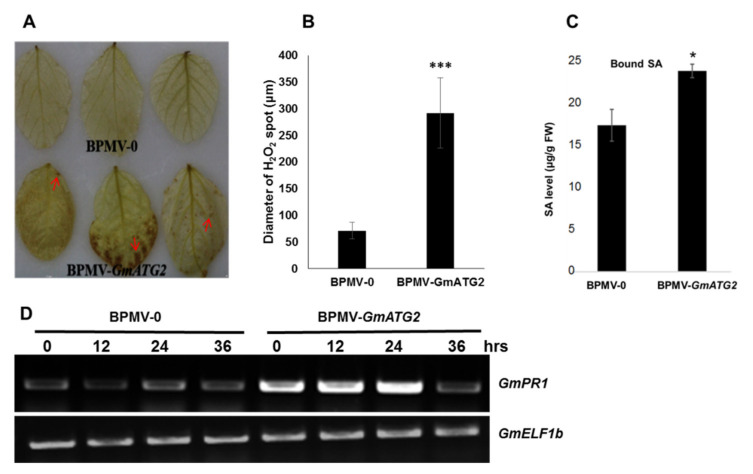
Defense responses are activated in *GmATG2*-silenced plants under dark conditions. (**A**) H_2_O_2_ level enhanced significantly in the BPMV-*GmATG2* plants relative to BPMV-0 plants treated in the dark for 3–4 days. The red arrows pointed to the sites of the ROS accumulation; (**B**) the diameters of H_2_O_2_ staining spots in **A** were measured under a dissecting microscopy. The data are shown as means ± SD. *** *p* < 0.001, Student’s *t*-test; (**C**) The bound SA level enhanced significantly in the BPMV-*GmATG2* plants relative to BPMV-0 plants. * *p* < 0.05, Student’s *t*-test; (**D**) Expression of *PR1* gene was induced to a much higher level in *GmATG2*-silenced plants than in vector control plants. RNA samples were extracted from the indicated plants at 0 min, 12 h, 24 h and 36 h post dark treatment; *GmELF1b* was used as the internal reference gene. These experiments were repeated at least twice with a similar result. At least 3 plants were used in each repeat.

**Figure 5 ijms-22-11749-f005:**
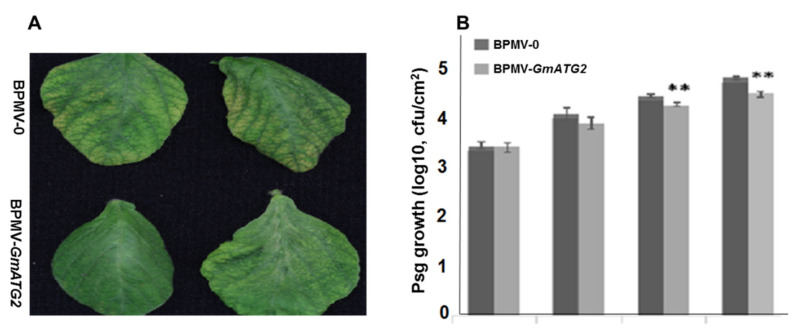
Silencing *GmATG2* results in enhanced resistance of soybean plants to *Pseudomonas syringae pv.* glycinea (*Psg*). (**A**) Comparison of the symptoms on the BPMV-*ATG2* plants and BPMV-0 plants caused by *Psg* infection at 9 days post inoculation. (**B**) The *Psg* growth curves on the leaves of BPMV-0 plants and *GmATG2*-silenced plants at different days of post *Psg* inoculation (dpi). ** indicates significant difference at 0.01 level by Student’s *t*-test. This experiment was repeated at least 3 times with similar result. At least 3 plants were used in each repeat.

**Figure 6 ijms-22-11749-f006:**
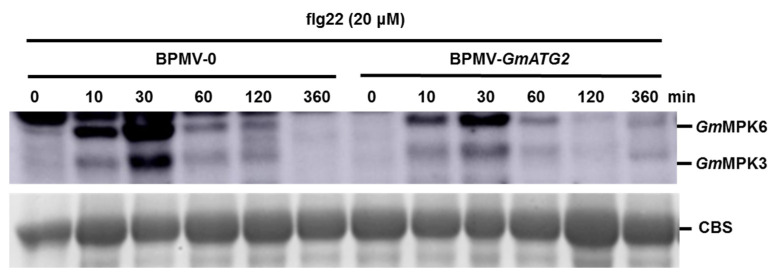
The activation of *Gm*MPK3/6 in response to flg22 is reduced in *GmATG2*-silenced plants relative to BPMV-0 plants. Leaf discs collected from BPMV-0 and BPMV-*GmATG2* plants were incubated on moist filter paper for 24 h to allow recovery from wounding before treatment with 10 µM flg22 for the indicated time. The kinase activities were detected by immunoblotting, using phospho-p44/42 MAP Erk1/2 antibody. Coomassie Blue staining (CBS) was performed for loading control. The experiment was repeated twice with a similar result.

**Table 1 ijms-22-11749-t001:** Homology shared between Arabidopsis and soybean *ATG2* genes.

Genes	*AtATG2*	*GmATG2a*	*GmATG2b*
*AtATG2*	100%	62%	61%
*GmATG2a*	62%	100%	96%

## Data Availability

Not applicable.

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
