# Peer review of "Silencing *Autophagy-Related Gene 2* (*ATG2*) Results in Accelerated Senescence and Enhanced Immunity in Soybean"

_ijms, 2021, doi:10.3390/ijms222111749_

Round 1
Reviewer 1 Report
This paper discusses the mechanism behind the autophagy in soybean and uses a transgenic model in which one of the autophagy-controlling genes has been silenced. The work describes a phenotype exhibiting accelerated senescence and pathogen resistance, independent of the MAPK signaling pathway. The study is interesting, and the manuscript is well written. There is one point that would still require clarification for the readers and that is what is the relevance of the exclusive observation of the accelerated senescence phenotype in the dark. How do light/dark conditions factor into the pathways investigated in this study?
Author Response
Responses to reviewer 1:
First of all, we would like to thank this reviewer for the positive comments on our manuscript!
What is the relevance of the exclusive observation of the accelerated senescence in the dark? How do dark/light conditions factor into the pathways investigated in this study?
Autophagy pathway functions to break down the organic molecules and replenish the carbon source for cells. Autophagy occurs at basal level and plays a role under normal growth conditions. However, autophagy is highly induced under various stress conditions especially under carbon or nitrogen starvation conditions. Because photosynthesis halts in the dark, dark treatment for a longer period of time mimics carbon starvation and therefore autophagy is greatly induced under dark conditions. Arabidopsis knockout mutants such as atg2, atg5 and atg7 display accelerated senescence phenotype even under normal growth conditions because of lacking functional autophagy. In the case of GmATG2-silenced plants, because GmATG2 was not completely silenced, certain level of autophagy is still operating in these plants, which is sufficient under normal growth conditions. However, the GmATG2 level in the GmATG2-silenced plants was not sufficient to maintain the autophagy flux under dark conditions (carbon starvation). That is the reason why the accelerated senescence was only exclusively observed in the GmATG2-silenced plants under dark conditions. We discussed this in the first paragraph of DISCUSSION section at page 14. The changed part is shown in red.
Reviewer 2 Report
The manuscript of Nashimi et al. deciphers the role of ATG2 gene in maintenance of senescence and immunity of soybean plants. The authors obtained mutant plants with silenced two copies of GmATG2 gene and compare its phenotype with empty vector control plants. Many similarities with phenotypes of autophagy mutants in Arabidopsis thaliana were observed.
According to presented research I have few main suggestions for authors.
- Analysis of developmental /age-dependent senescence in transgenic soybean plants should be performed since some early senescence symptoms can be already visible on BPMV-GmATG2 plants presented at Fig. 1A. Moreover, this result should give very important information about the role of GmATG2 in plants in non-stress conditions.
- The information about number of repetitions of each experiment, sample sizes etc. is missing.
- Please show the blast results for GmATG2s and AtATG2.
- The text description of Fig. 1C and D is inconsistent with the Fig. 1 legend and picture.
- Fig. 3A – the result is rather weakly convincing. Please perform the densitometry analysis of obtained result or show better picture or use another method to show enhanced ATG8 accumulation in BPMV-GmATG2 plants.
- In lines 310-312 Authors stated that “examining the formation of autophagosomes by using a GFP-GmATG8 fusion protein are required to draw a firm conclusion whether the autophagy pathway is indeed compromised in the GmATG2-silenced plants”. That is the basic and obligatory experiment to prove that silencing of GmATG2 in fact disturbed autophagy in soybean and all observed phenotypes are consequence of that. Please add this data.
- Please discuss better the result of MAPK activation by treatment of GmATG2 –silenced plants with flg22.
- Line 382 – the results showing cell death are not presented in the manuscript. Please add the data.
Minor issues:
- The name of company providing plant antibodies is Agrisera – please correct multiple times in the manuscript.
- Line 33 – malnutrition is used rather in a medical meaning, please use another formulation like “nutrient deprivation” or other.
- Fig. 4 – Please write how long plants were treated by darkness.
Author Response
Responses to reviewer 2:
Major points:
- Analysis of developmental/age-dependent senescence in transgenic soybean plants should be performed since some early senescence symptoms can be already visible on BPMV-GmATG2 plants presented at Fig. 1A. Moreover, this result should give very important information about the role of the GmATG2 in plants in non-stress conditions.
We totally agree with this reviewer’s suggestion. Generation of transgenic plants is always the best approach for gene function studies in plants. However, transformation in soybean is extremely difficult and only a few labs can do so. We once had a master student working on soybean transformation project following an optimized protocol from an established lab. Despite of great efforts, she did not even generate a single transgenic soybean plant. For this reason, we cannot provide this data.
- The information about number of repetitions of each experiment, sample size ect is missing.
Thank this reviewer for pointing this out! We have added this information in the figure legends.
- Please show the blast results for GmATG2s and At
We provided this information in Table 1.
- The test description of Fig. 1C and D is inconsistent with the Fig. 1 legend and picture.
Thank this reviewer for catching the errors. We have switched the position of Fig. 1C and 1D and made the corresponding changes in Figure legend.
- 3A—the result is rather weakly convincing. Please perform the densitometry analysis of obtained result or show better picture or use another method to show enhanced ATG8 accumulation in BPMV-GmATG2 plants.
Following the reviewer’s suggestion, we have replaced the original image with a new one. We would like to point out that the new image was actually the very first Western blotting we performed. Unfortunately, the band of 36-hr sample form the silenced plant was missing in this new figure due to a technique error. Nonetheless, this new Fig. 3A is much more convincing than the original one in our initial submission.
- In line 310-312, authors stated that “examining the formation of autophagosomes by using a GFP-GmATG8 fusion protein are required to draw a firm conclusion whether the autophagy pathway is indeed compromised in the GmATG2-silenced plants.” That is the basic and obligatory experiment to prove the silencing and observed phenotypes are consequence of that. Please add this data.
Thank this reviewer for bringing this up! We are fully aware of this and thought about doing this experiment. That’s why we discussed this in our manuscript. However, we encountered dilemma situations that prevented us from achieving this goal. We used VIGS approach to generate GmATG2-silenced plants. By the time we obtained the silenced plants, the plants were already 3-4 weeks old. It is well known that the mature soybean leaves have heavy and thick hairs, which severely interferes with the observation of the leaf cells under Confocal microscope. In addition, unlike in N. benthamiana or Arabidopsis, transient expression of GFP-ATG8 construct in soybean leaves via agro-infiltration is extremely difficult. With these two obstacles in soybean community, we had no choice but drop the idea. If we could find collaborators or have funding for generating transgenic soybean plants commercially, we would perform this experiment in the GmATG2-silenced or knockout plants in the future.
- Please discuss better the result of MAPK activation by treatment of GmATG2-silenced plants with flg22.
Following the reviewer’s suggestion, we have discussed more regarding the MAPK activation by flg22 (see the changed part in red at page 16).
- Line 382- the results showing cell death are not presented in the manuscript. Please add the data.
Thank the reviewer for catching this error. We mistakenly added this sentence. We have deleted it.
Minor issues:
- The name of company providing antibody is Agrisera- please correct multiple times in the manuscript.
Thank the reviewer for catching the mistakes. We have made the changes.
- Line 33- malnutrition is used rather in a medical meaning, please use another formulation like “nutrient deprivation” or other.
Thank the reviewer for correcting the term usage. We have changed to “nutrient deprivation” as the reviewer suggested.
- Please write how long plants were treated by darkness.
The plants were treated under the dark for 3-4 days. We have added this information in the Results section as well as in the legend of Figure 4.
Round 2
Reviewer 2 Report
Dear Authors,
thank you for all your comments and corrections. I appreciate them.
Please add "Anti-ATG8" mark at new Fig. 3.